# Participation of Electric Vehicle Aggregators in Ancillary Services Considering Users' Preferences

**Jean-Michel Clairand** 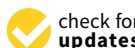

Facultad de Ingeniería y Ciencias Agropecuarias, Universidad de las Américas-Ecuador, Quito 170122, Ecuador; jean.clairand@udla.edu.ec; Tel.: +593-9-95860613

**Abstract:** Growing environmental concerns have contributed to urban transit alternatives, such as Electric Vehicles (EVS). As a result, the market for EVs is growing each year, which are a solution to mitigate these concerns. Although EVs present several environmental advantages, a massive introduction of them could generate power systems issues. Several works have proposed strategies to mitigate those issues. Since EVs posses batteries with significant capacity, they could provide services to the power grid, such as ancillary services. Thus, this paper presents a methodology where EVs could participate in Secondary frequency response through an EV aggregator. Moreover, EV user's preferences are taken into account to facilitate their participation. The case study of Quito, Ecuador is considered. The results of this methodology show that the EV aggregator has a significant potential for providing electricity market regulation services, especially in view of the use of V2G mode.

**Keywords:** aggregator; ancillary services; electric vehcicle; smart grid; smart charging

## 1. Introduction

In order to reduce greenhouse gas emission and regional emissions in towns, electric vehicles (EVs) are seen as efficient transportation technologies. Many governments have taken steps to encourage the purchase of EVs for private clients. Nonetheless, a large deployment of EVs can lead to power grid problems, including voltage falls [1], major distribution investments [2], and power losses [3]. Therefore, the capacity of hosting distribution networks must be evaluated to ensure the correct grid constraints. In the future, EV users will become active prosumers instead of passive customers who can supply electricity services such as ancillary utilities, such as [4].

Several researchers have studied new methodologies to mitigate grid issues due to this new significant load. These methodologies include peak reductions, cost reductions, power losses reductions, among others [3,5–7].

In addition to these objectives, it is possible to take advantage of the EV flexibility in the charging process (or even to consider the possibility do discharge them) to provide ancillary services such as frequency control. EVs could participate in primary, secondary and tertiary response. Thus, some other works have studied the participation of EVs in ancillary services. For example, in [8], an optimal scheduling of V2G energy and ancillary services is discussed. In [9], an online scheduling method that does depend on the forecast of the regulation demand and allows each EV to determine its own schedule in real time, is proposed to provide V2G services. In [10], a multi-market-driven microgrid energy schedule including distributed and centralized market participation, which closed the gap between the internal ancillary services market and external wholesale market, is studied.

In particular, several works have considered the participation of EV aggregators that interact between the charging of EVs and power systems. The authors of [11] present the stochastic scheduling of aggregators of plug-in electric vehicles for participation in energy and ancillary service markets. The authors of [12] propose an optimal dispatching strategy for V2G aggregator participating in

supplementary frequency regulation considering EV driving demand and aggregator's benefit. In [13], an EV aggregator bidding strategy in the day- ahead market is proposed, considering reserve capacity and deployment. The authors of [14] discussed a decentralized EV charging control strategy of the EV aggregator for scheduling the flexible charging demand of PHEVs in residential distribution networks. In [15], a robust optimization technique is used for the robust scheduling of EV aggregators considering price uncertainty. The authors of [16] propose a bi-level stochastic optimization model of offering strategy for an aggregated wind power producers-EV hybrid power plant as a price maker in the day-ahead market considering the uncertainties of the energy production and spot price in the real-time market. In [17], an optimal participation of an EV aggregator in energy market with uncertain prices was investigated by robust optimization method. The authors of [18] studied an interval optimization approach to model upstream grid price uncertainty.

Although these works and others present innovative solutions for the participation of EVs in ancillary services, no work offers any advantage for EV users flexibility. Thus, the aim of this paper is to present a methodology for participation of EVs in secondary frequency response but considering different user's preferences. A previous work [19] discussed the charging minimization costs of EVs considering user's preferences, by customer choice products (CCPs) in terms of charging time, but did not consider the participation in ancillary services. Hence, in this work, the EV aggregator should also adjust slow charging power to fulfill technical constraints imposed by network operators while EVs are charged at the lowest cost, but also, it should provide regulation reserves to the power grid. The EV users should consider the same CCPs than [19], but the charging process will be different and is detailed in the methodology. The main contributions of this works are highlighted as follows:

- The EV aggregator will be in charge of avoiding technical issues in the electricity network, while the charging costs and regulation services are optimized, both in the operational planning and in real-time.
- A charging and discharging management of the EV fleet is studied for determining if V2G is suitable or not for regulation services.
- The EV aggregator potential profitability from providing regulation services is studied.

The rest of this paper is organized as follows. Section 2 presents background of ancillary services. Section 3 presents the methodology formulation. Then, the different inputs of the case study are detailed in Section 4. The results are discussed in Section 5. Finally, Section 6 is devoted to conclusions.

## 2. Background: Ancillary Services

### 2.1. Control as Ancillary Services

Several services related to power generation, transmission and distribution were separated and offered by various actors through the liberalization of the electricity markets. For instance, ancillary services are defined as the services needed to support the reliable transmission of the grid from suppliers to users, such as frequency and voltage regulation [20].

On the one hand, competitive markets for ancillary services are being designed, especially for frequency control, given the way that these services can be straightforwardly linked with active power and energy production, and henceforth can be promptly priced. On the other hand, markets for voltage control services are as of now immature in good part because these services are related to reactive power (MVar), which is more difficult to trade as the effect of reactive power in the voltage is more local [21].

In fact, although frequency control markets that are also mentioned as ancillary service markets do not have a single market structure in most jurisdictions to provide frequency controls, they rely on the bidding and contracting instruments for buying spinning and non spinning frequency control reserves for both generators and loads [22].

## 2.2. Frequency Control

For satisfactory operation of a power grid, the frequency ought to remain nearly constant. Generally, close control of frequency guarantees steadiness of speed and induction and synchronous motors. The frequency of a system is subject to active power balance. A variation in active power demand at one point leads to a difference in frequency. Since there are several generators providing power into the system, a few capabilities must be given to allocating change in demand to the generators. The primary control function is given by a speed governor A speed governor on each generating unit gives the primary speed control function. In an interconnected system with at least two independently controlled areas, in addition to control or frequency, the generation within each area must be controlled in order to keep scheduled power exchange. The control of generation and frequency is commonly referred to as load-frequency control (LFC) [23].

Moreover, power system control is implemented for both frequency and voltage in three steps: primary, secondary, and tertiary. These steps have different objectives, time response, geographical conditions, but they are always in interaction between them. Firstly, the primary response is based on the local control performed by the generator, in order to stabilize the system frequency if a disturbance occurs. The response takes some seconds and it is not responsible for restoring the reference value of the system frequency. Then, the secondary control is associated with preserving the power balance in a control area, as well as keeping the system frequency. The response takes several seconds to minutes. Finally, the tertiary control is responsible for modifying the active power set points in the generators in order to perform a global power system operating strategy. The response takes several minutes [21]. Hence, it is crucial to have generation reserves to account for abnormal conditions. Considering the time responses of the different reserves, the EV load can act especially as a resource for secondary reserves, such as presented in [24–26].

## 2.3. Regulation Services and EVs

Regulation has to be under direct real-time grid operator control, which enables the generator unit to receive and react within a minute or less by increasing or decreasing the generator output from the computer of the grid operator [25]. Regulation can be divided into two classes regulation up and regulation down. Regulation up is the ability to increase power generation from a prediction level and regulation down is to decrease power generation. For example, if load exceeds generation, the frequency will drop and an additional generation power is required, so in this case regulation up is required. In particular, the market can request both for regulation up and regulation down. The regulation calls for the generators are significant (more than 400 per day), so the energy offers are each day or even each hour and they are based on the purchase or sale of a quantity of electricity. Then the market matches offers and bids to determine the price and the amount of electricity that each one is awarded. Furthermore, the response of the power available from the EVs has to be quick for the participation in regulation services due to the possible significant calls. Hence, the authors of [25] also demonstrated such feasibility.

A few works of EV participation in ancillary services exist in the literature compared to other fields of study. But, none of them offer to the EV customers the possibility to charge according to their preferences at the moment they need to charge. For this reason, the aim of this paper is to give an approach based on the CCPs defined in [19]. Moreover, in [27], it was demonstrated that the V2G mode offers small additional benefits more than the ones that can be achieved with modulating the charge (as proposed in this thesis), with significantly smaller cost (in control equipment and battery degradation). This paper discusses the possible benefits with the inclusion of V2G mode in the green and blue CCP.

## 3. Methodology Formulation

### 3.1. Optimization Formulation

In [19], the minimization of the charging costs was considered, which is performed by the EV aggregator. Nevertheless, the benefits of the EV aggregator could be improved with the participation of it in the regulation services. Thus, in addition of taking advantages of time-varying electricity CCPs by charging the EVs, the EV aggregator should participate offering operational reserves during the operation horizon.

Based on the previous methodology of smart charging, a new methodology is developed, which maximizes the EV aggregator benefits considering ancillary services. It is based on an optimization formulation that is subject to different constraints. In addition, the methodology considers the study with and without the V2G mode. Depending on its experience, it could earn more benefits making higher regulation bids. The EV aggregators benefits are defined:

$$B = R - C \tag{1}$$

where the EV aggregator revenues are defined as follows:

$$R = E^D \times \pi^{E,D} + E^U \times \pi^{E,U} + \sum_{t=1}^{T} [R_t^D \times c_t^D + R_t^U \times c_t^U] \times \Delta T \tag{2}$$

With:
$\pi^{E,D}$: Market selling price for regulation down ($/kWh)
$\pi^{E,U}$: Market selling price for regulation up ($/kWh)
$R_t^U$: Regulation capacity up available at time $t$ (kW)
$R_t^D$: Regulation capacity down available at time $t$ (kW)
$c_t^D$: Hourly regulation capacity price down ($/kW-h)
$c_t^D$: Hourly regulation capacity price up ($/kW-h)
$E^D$: Dispatched energy for regulation down (kWh)
$E^U$: Dispatched energy for regulation up (kWh)

The revenues include a first component for energy payment ($/kWh), and the other component for availability ($/kW-h), both of them for regulation up and down. The first term is considered the energy revenue and the second the capacity revenue. Note that the energy component corresponds to a revenue of the EV aggregator if it is matched by the regulation market, and the capacity payment is due for the power available.

The costs include a cost of energy, a penalty charge, an associated cost from battery degradation associated from discharging the EV battery (in case V2G is not used, this cost has not to be included), a cost for the bidirectional chargers, and a payment for the users; they are defined as follows:

$$C = C_E + C_P + C_B + C_H + C_U \tag{3}$$

where:

$$C_E = \sum_{t=1}^{T} P_t^{EV} \times c_t^e \times \Delta T \tag{4}$$

$$C_P = \alpha_P \times E_P \tag{5}$$

$$C_B = \sum_{t=1}^{T} \alpha_{bd} \times P_t^{EV,d} \times \Delta T \tag{6}$$

$$C_H = \alpha_H \times D \times (N^G + N^B) \tag{7}$$

$$C_U = \alpha_U \times R \tag{8}$$

In particular, Equation (5) indicates that if the EV aggregator fails to comply with the given regulation bid in the real-time, a penalty will be imposed at the end of the current day. A penalty cost was assumed considering a penalty coefficient for the energy that was not consumed or delivered. Equation (6) indicates that the EV aggregator has to discount an amount for the battery degradation that is considered for the V2G mode when the EV supply energy to the grid. This penalty cost was not considered for the G2V mode since it is minimal compared to V2G. The parameter $\alpha_{bd}$ is a factor that defines the battery degradation per kWh absorbed from the EV battery. Equation (7) corresponds to the cost for the investment of bidirectional chargers, such as indicated in [28]. Since the cost was assumed in this work for 390 $ for the installation of a bidirectional charger, it is considered that the EV aggregator has to pay a cost per day and per EV charging, so the factor for the bidirectional charger was assumed to be $\alpha_B$ = 0.2 $ per day and per charging station. Note that only green and blue CCP users are considered for these costs, they're the only one that bidirectional chargers can use. Equation (8) indicates that a part of the revenues from the EV aggregator has to be distributed to the EV users. In this way, EV users could be interested for participating in ancillary services. It is assumed $\alpha_U$ = 5%.

Observe that in this case, the revenues are not constant anymore and depends on the capacity of the EV aggregator for offering regulation services. This is because in this case the revenues depend on the regulation bids, which vary during the day, so the revenues can be maximized in the day horizon.

The EV aggregator has to set an hourly prediction load profile so it can provide the frequency regulation service by varying its charging loads above or below the prediction. Under this mechanism, the default status of all the EVs at any decision stage is "charge" [29]. The prediction of the EV aggregator is defined:

$$P_t^{bas} = \sum_{i=1}^{N} P_{i,t} \tag{9}$$

Also, each whole EV CCP load is calculated as, for x $\in$ {green, blue, red}:

$$P_k^x = \sum_{i=1}^{N_x} P_{x,k,i} \tag{10}$$

The objective function can be written as follows:

$$maxB = maxR - C \tag{11}$$

Observe that the objective function is developed based on Equations (2) and (3).
The methodology is subject to the different constraints:

$$\underline{P^{EV}} \leq P_{i,t} \leq \overline{P_{max}^{EV}} \tag{12}$$

$$\underline{SOC^{EV}} \leq SOC_{i,t} \leq \overline{SOC^{EV}} \tag{13}$$

$$R_t^{D,B} \leq R_t^D \tag{14}$$

$$R_t^{U,B} \leq R_t^U \tag{15}$$

With:
$R_t^{D,B}$: Offered regulation bid down for time $t$ (%)
$R_t^{U,B}$: Offered regulation bid up for time $t$ (%)

Equation (12) indicates the limits of the power supplied or demanded from an EV charger. Equation (13) defines the SOC limits for a better operation of the EV battery for mitigating the effects of degradation. Equations (14) and (15) ensure that the regulation bids for the day ahead are at their minimum available. In some works, the minimum reserve is assumed to be 0.1 MW for participating in regulation services, considering the values of PJM Market [28,30,31]. But, since the power levels of the Ecuadorean grid are smaller, a value of 50 kW is considered. In addition, the offered regulation bids are assumed to be 50% of the EV availbale load. Depending on the performance of the EV aggregator, it can be increased for receiving higher benefits.

### 3.2. Aggregate Model

For the ancillary services participation, the EV aggregator needs to sum the energy and power boundaries of all the EVs, depending on the CCPs the owners request. In particular, EV users who select green or blue CCP are able to participate both in regulation down and up, but red CCP users are only able to participate in regulation down, because they do not allow flexibility for decreasing the charging power rate. Observe, that red CCP do not participate in V2G mode but also their users charge at maximum power.

The aggregate charging power of all the EVs can be calculated as follows:

$$P_t^{EV} = \sum_{i=1}^{N_{EV}} P_{i,t}^{EV} = \sum_{i=1}^{N_{G,EV}} P_{i,t}^{G} + \sum_{i=1}^{N_{B,EV}} P_{i,t}^{B} + \sum_{i=1}^{N_{R,EV}} P_{i,t}^{R} \tag{16}$$

The additional constraints for the CCPs are listed as follows:

$$R_t^{D,B} \leq \sum_{i=1}^{N_{G,EV}} P_{i,t}^{G} + \sum_{i=1}^{N_{B,EV}} P_{i,t}^{B} + \sum_{i=1}^{N_{R,EV}} P_{i,t}^{R} \tag{17}$$

$$R_t^{U,B} \leq \sum_{i=1}^{N_{G,EV}} P_{i,t}^{G} + \sum_{i=1}^{N_{B,EV}} P_{i,t}^{B} \tag{18}$$

Equations (17) and (18) indicate that all the power available for regulation has to be higher than the offered regulation bids. The constraint is performed based on the forecast of the day before. At the end of the day, the EV aggregator has to pay the penalty cost for the energy that fail to comply with the regulation bids.

### 3.3. Charging Prices

When an EV $i$ is plugged the EV aggregator calculates the corresponding price to pay that is defined:

$$p_i = \sum_{k=1}^{k=D} \pi_k \times P_{k,i} + p^F - \frac{C_U}{N}, \forall k \in U_i \tag{19}$$

The EV user has to pay the variable electricity price and a constant value $p^F$ that corresponds to the benefits of the EV aggregator, which corresponds to its revenues for the service provided. The EV user has a discount corresponding to the participation in regulation services, which corresponds to the last term and comes from the cost of the EV aggregator from Equation (19). Note that the price is calculated for each time horizon $U_i$, which is different for all users.

### 3.4. Forecast and Real-Time Implementation

The methodology is divided in two time parts:

- The first considers the day-ahead optimization considering the prices for the next day of the regulation up and down, and electricity prices, and considering the forecast of the EV users behavior, such as hour of arrival, energy required, and CCP selected.

- The second time considers the real-time optimization. The model is optimized in real time, considering the new arrivals and the updated prices of regulation up and down and electricity prices. For the real-time optimization, the different values are updated each 15 min.

The methodology should guarantee a high accuracy and the maximum exploitation of the potential regulation services.

The different steps are considered for this methodology:

- At the end of the previous day to the day in which the charging process is going to be scheduled, the EV aggregator receives from the electricity market the predictions for the next day of the electricity price and regulation services requirements. Furthermore, the EV aggregator has to build the EV load prediction curve, based of the forecast of the charge of the EVs that is obtained from the information of the arrivals and energy required from the EV users from the current and previous day. Then it performs the different regulation offers and send to the Operator.
- At the start of the new day, it has to be considered EV charging that was not completed the ending day. This is a result available from the optimization performed for this day.
- The aggregator must receive information on new plugged cars in real-time at the beginning of every 15 min. The process is performed using smart meters at the customer's facilities, where EV users will receive in real-time information regarding the costs of the various CCPs. If new cars are connected for loading, each smart meter has to send the associated information to the aggregator: State of Charge and CCP selected. The aggregator will combine this information with the information about the actual and short term EV status.
- An optimization process is performed to obtain a charging profile for each EV taking into account the calculated charging period $U_i$ and the total power needed. As a result, the moment when an EV will be totally charged will be known. The optimization is performed for each EV connected in a specific moment and not for the sum all of them. The dispatch ratio is used to calculate the dispatched energy. The optimization process will determine the new charging according to both the network constraints and the committed charging in previous steps. The optimization process takes into account the possible energy calls. In a case that in real-time, there is not enough energy available, the EV aggregator will pay the penalty cost, instead of not satisfying the EV users requirements.
- The phase will be repeated until the end of the next day every 15 min. Optimization will take into account the continuation in the next day of EV charging if charge start late in the day before, but this pattern of charge has to be taken into account at the start of the new day.
- At the end of the day, the benefits of the EV aggregator are calculated.

The flowchart of the proposed methodology is shown in Figure 1.

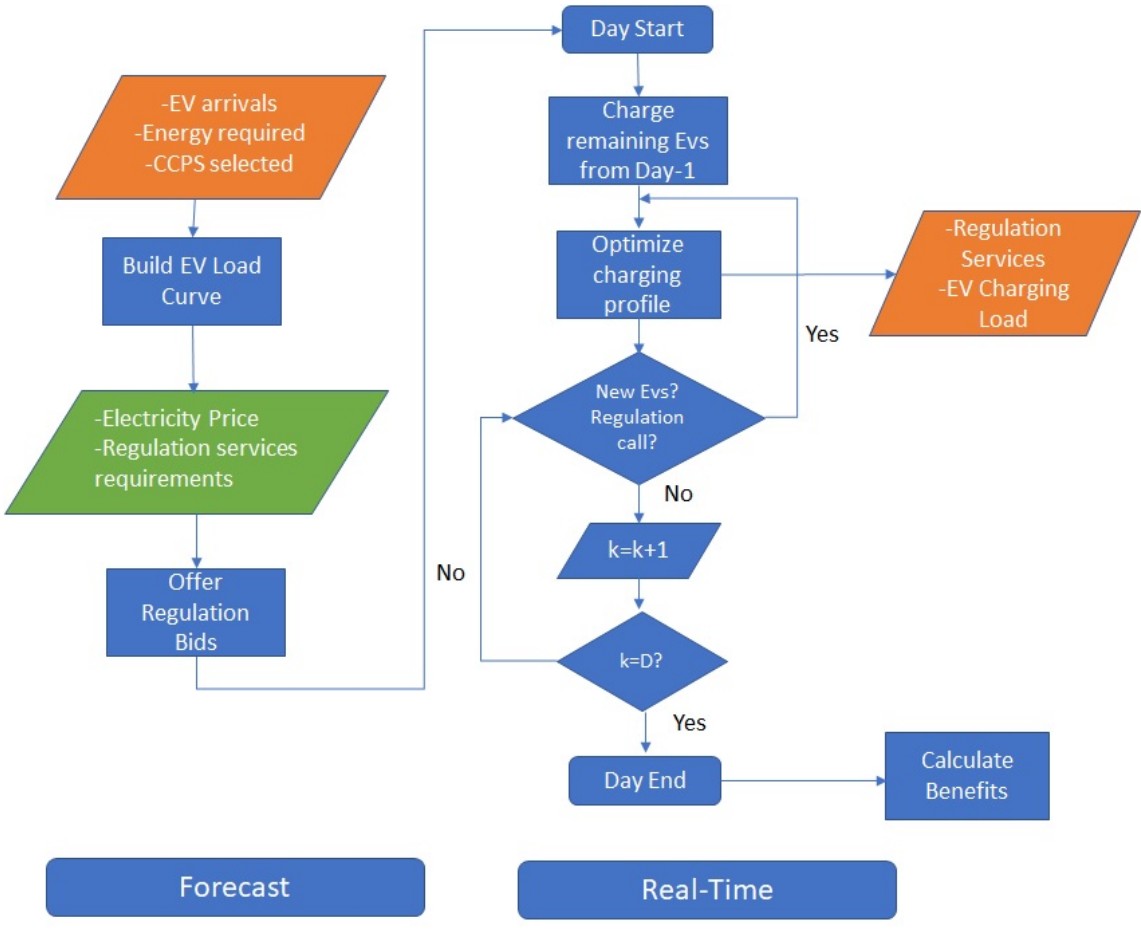

**Figure 1.** Flowchart of regulation participation.

## 4. Case Study

A case study is presented that demonstrates the effectiveness of the proposed methodology. The data from the traffic of Quito, Ecuador is considered.

### 4.1. EV Input Variables

#### 4.1.1. Number of EVs of Each CCP

In a previous work [32], the number of vehicles in this area was concluded to be almost 1000. The total number of vehicles is assumed to be this value.

In the case study evaluation, the green, blue and red portions of the CCPs are assumed in this study to be 60%, 30% and 10%. In next section, the sensitivity analysis of these values is performed. The Table 1 describes the number of vehicle for each penetration level of EV.

**Table 1.** Number of vehicles from each CCP depending on penetration level.

| Penetration Level | $N^G$ | $N^B$ | $N^R$ |
|---|---|---|---|
| 50% | 300 | 150 | 50 |
| 75% | 450 | 225 | 75 |
| 100% | 600 | 300 | 100 |

### 4.1.2. Starting Charging Time

Studies on road transport in Quito and working conditions find the beginning of charge times as follows [33,34]:

- A 20% of EV users who participate in the program plug their EV at work between 07:00 and 10:30.
- A 40% of EV users who participate in the program plug their EV at home, after returning from work between 16:00 and 21:00.
- The rest of EV users who participate in the program plug their EV in different periods of the day (shops, home, work, etc.).

Random numbers will be created according to these schedules to create starting charging time profiles. For each vehicle, a starting hour $st^i$ is assigned, where i $\in \mathbb{N}$. Figure 2 shows the histogram corresponding to 500 EV users' starting time in a day.

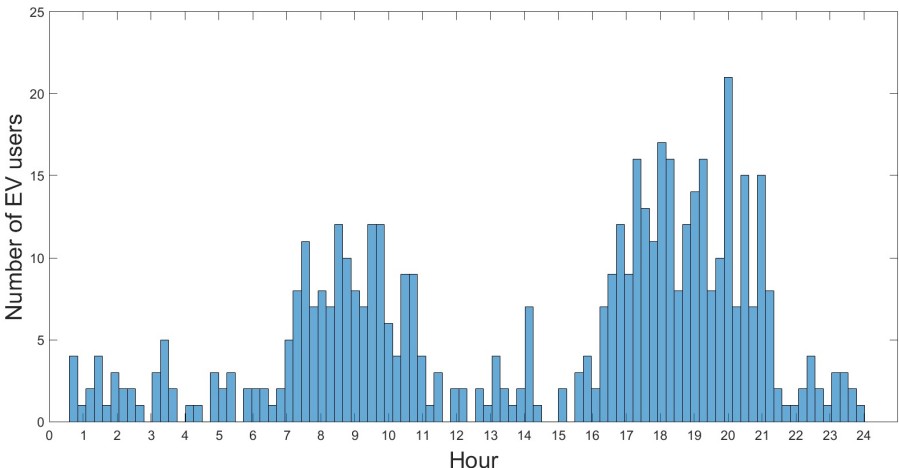

**Figure 2.** Histogram of the starting charging time of EV users in a day in the selected zone in Quito [19].

### 4.1.3. Daily Energy Needed from Each EV

Various types of EVs (brands and models) are expected to be seen shortly on roads in Ecuador. Nissan Leaf, BYD e5, BYD e6 and Kia Soul EV are amongst these. These EVs have different battery capacities from 24 kWh to 75 kWh of Nissan Leaf and BYD e6 respectively. For that reason, it is more valuable to consider in the calculation the daily energy demanded by each EV user than the battery capacities of each EV. For this study, it is assumed to follow the pattern of that probability curve with different levels of energy, but considering last average value . Moreover, it is established, that EV users must charge at least 4 kWh in order to participate in this EV aggregator program. It is considered that the more suitable curve is Weibull density probability, from where random values will be selected [35]. This curve is defined as:

$$f(x; a, b, c) = \frac{b}{a} \times (\frac{x+c}{a})^{b-1} . e^{-(\frac{x+c}{a})^b} \qquad (20)$$

The charging needs may vary from 4 kWh to 28 kWh, but with more values between 6 and 10 kWh, that can be normal values of energy required [36]. This curve allows to have different values of energy required by user, that differs importantly. It was also mentioned that models as BYD e6 have a battery capacity of 60 kWh, but the case of users that have this EVs and charge more than 28 kWh are ignored. Note that these parameters are chosen with the behavioral conditions of Quito individuals, but this curve could differ significantly in other areas. So, the values for the parameters selected are: a = 7.5; b = 1.5; c = 4. Density probability curve is obtained in Figure 3. Note that the integral of this curve is equal to 1.

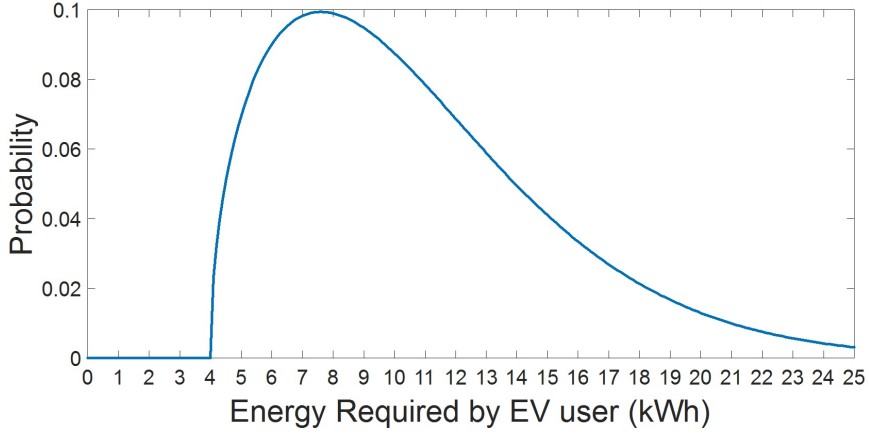

**Figure 3.** Probability Density Curve of Energy required by EV user [19].

It is considered that EV users consumed energy is a random number from this distribution.

From this way, each set of time intervals $U_i$ that correspond to charging period of an EV *i* is defined by:

$$U_i = [st_i; st_i + \frac{E^{req}}{P^{av}}] \tag{21}$$

### 4.2. CCPs Average Charging Power

Each user needs to choose a CCP according to the Table 2, that indicates average charging power Table 2. Slow charging could range from zero to 7.2 kW, as previously mentioned. The maximum value is to be chosen for Red CCP, and small values are to be chosen for the other two CCPs to optimize the time. These values were selected for the case study:

**Table 2.** Average power for each CCP.

| CCP | Green | Blue | Red |
|---|---|---|---|
| $P_{x,av}$ (kW) | 1.5 | 2.5 | 7.2 |

These values were selected in order to have intermediate values in the range of slow charging power rates that are 0–7.2 kW.

Actually, EV users who select red CCP will have its charging power rate constant and established at its maximum value. There will not be an optimization for this case. Users assume the price for this condition.

### 4.3. Ancillary Services Data

Ecuador does not have an electricity ancillary services market, especially for regulation services. Moreover, Ecuador has to pay those services to foreign countries [32]. The national operator of electricity CENACE has to manage the frequency control. The secondary control is carried out through the generating units assigned by CENACE for this purpose, which are generally thermal and hydroelectric [37].

For this reason the data from ESIOS of Red Eléctrica España is selected, for validation purposes [38]. These data are published resulting from the operation processes of the electrical system that are the responsibility of Red Eléctrica España (REE) as System Operator. The information is published as soon as it is generated in the operating system. Each process has its periodicity (forecasts, programming, receipt of measures, liquidation, among others). Note that this information is added as an input, so as if the market existed in Ecuador, it could be considered for this methodology. A typical day of the week is selected for the study. Figure 4 represent the price profiles for regulation up

and Figure 5 the price profiles for regulation down. Note that the forecast and real-time prices are represented. The battery degradation cost $\alpha_{bd}$ was fixed at 0.004 \$/%-h according to [39] and the penalty factors $\alpha_p$ at 0.004 \$/kWh according to [29]. Moreover, the energy revenue could be defined by dispatch ratios defined:

$$r^D = \frac{E^D}{P_t^{EV} \times U_i} \tag{22}$$

$$r^U = \frac{E^U}{P_t^{EV} \times U_i} \tag{23}$$

The energy dispatched for regulation services is some fraction of the total power available and contracted. For this reason, this ratio is used [25]. The values from [40] are considered: 0.033 and 0.062 for regulation up and down respectively.

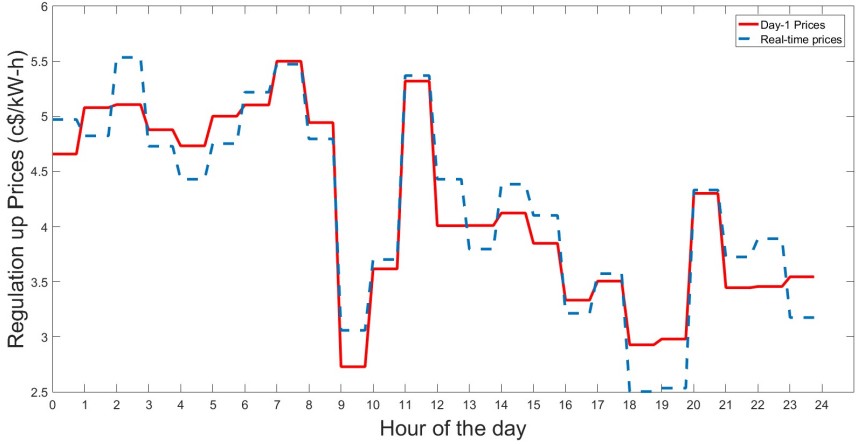

**Figure 4.** Regulation Up Prices.

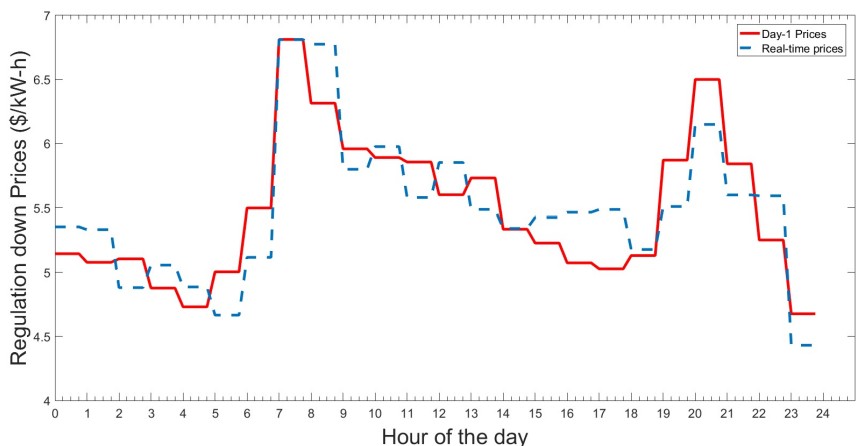

**Figure 5.** Regulation down prices.

## 5. Results

### 5.1. Daily Operation

In Figure 6, the hourly regulation up for both in the forecast and in real-time are illustrated. Note that the variations between the real-time and the forecast are not significant. The peaks correspond to periods when the electricity is relatively expensive and the regulation up prices are high.

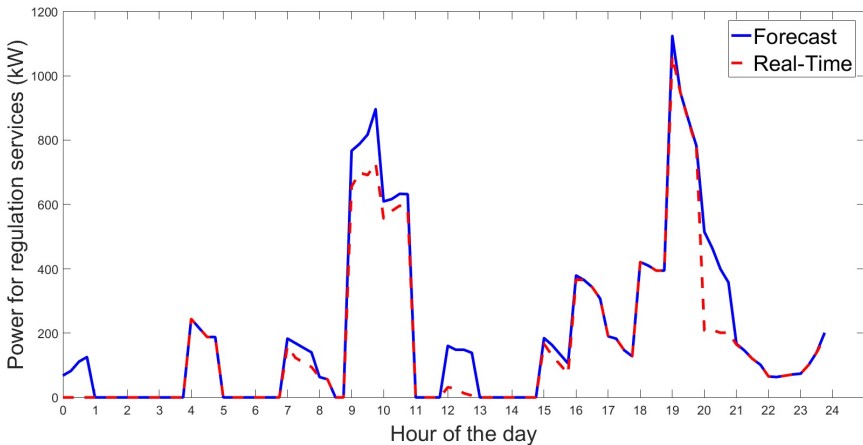

**Figure 6.** Up Regulation.

In Figure 7 the hourly regulation down for both in the forecast and in real-time are illustrated. Different variations during the day are observed, which correspond to the charging needs from the EV aggregator, which has to exploit at maximum the lower electricity prices and significant down regulation prices. Note that the EV aggregator can provide at the same time both regulation up and down. This is because the time arrival of the EVs are different, so the EV aggregator can increase the charging power rate of some of them and decrease others, depending on the problem conditions. Moreover, the points power of regulation that is below the curve of the forecasted correspond to the time periods that the EV aggregator cannot satisfy the regulation it offered the day before, so it has to pay the penalty cots corresponding to the energy difference.

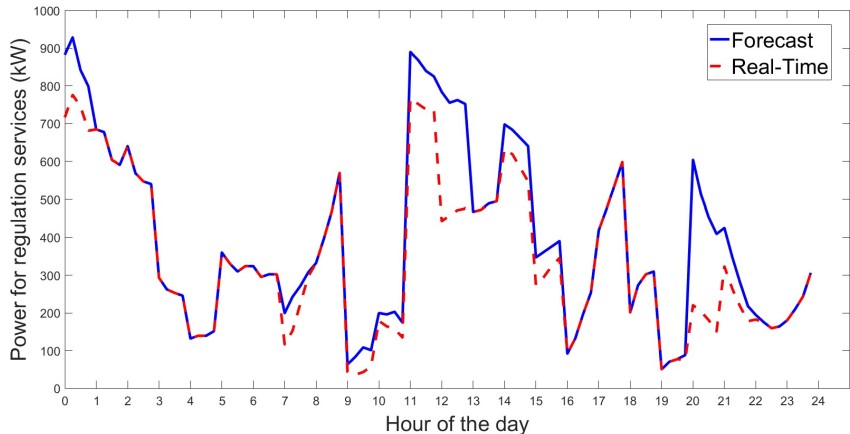

**Figure 7.** Down Regulation.

### 5.2. Daily Benefits

Results of the calculations of the daily benefits with V2G are shown in Table 3. Results of the calculations of the daily benefits without V2G are shown in Table 4. It is clear that regulation Down offers much more incomes to EV aggregator. In the case of the V2G mode, it is observed that the revenues are significant, but also the costs. In the case without the V2G mode, it is noted that the revenues for the regulation up are very small and for the regulation down they are not so significant. This can be explained due to the higher amount of power that the EV aggregator can manage with the V2G mode. Nevertheless, note that the indicated investments are indicative and can result more significant, and it is not sure that the EV users are willing to let their EV battery to discharge to the grid. Moreover, the regulation could be higher if EV users stay parked longer time than considered for the case study, but it is crucial to study the willingness of the EV users to remain parked a more extended period. Note that the number of EV users are 1000 because it corresponds just to a zone of

Quito. The benefits for the regulation services may be much higher in a whole city, and the regulation services more significant. Finally, the total benefits are not significant between the forecast and the real-time application, and the difference is mostly because of the penalty charge from the deviations of the regulation that is not provided by the EV aggregator. Thus, the methodology works correctly.

**Table 3.** Economic factors for EV participation in Regulation Markets considering the use of V2G.

|  | Forecast ($) | Real-Time ($) |
|---|---|---|
| Reg. Down Capacity | 485.3 | 501.0 |
| Reg.Up Capacity | 158.2 | 154.7 |
| Reg. Down Energy | 138.7 | 143.1 |
| Reg.Up Energy | 26.4 | 25.8 |
| Total Revenues | 808.6 | 801.6 |
| Energy | 333.7 | 335.2 |
| Penalty Charge | 0 | 7.2 |
| Bat. Deg. | 19.9 | 19.4 |
| Bidirectional electronics | 80 | 80 |
| Payment to users | 40.4 | 40.0 |
| Total Costs | 474.0 | 481.8 |
| Total Benefits | 334.6 | 319.8 |

**Table 4.** Economic factors for EV participation in Regulation Markets without V2G mode.

|  | Forecast ($) | Real-Time ($) |
|---|---|---|
| Reg. Down Capacity | 306.5 | 308.0 |
| Reg.Up Capacity | 24.7 | 29.0 |
| Reg. Down Energy | 87.5 | 88.0 |
| Reg.Up Energy | 4.1 | 4.8 |
| Total Revenues | 422.8 | 429.8 |
| Energy | 329.7 | 333.2 |
| Penalty Charge | 0 | 7.4 |
| Bat. Deg. | 0 | 0 |
| Payment to users | 21.1 | 21.5 |
| Total Costs | 350.8 | 362.1 |
| Total Benefits | 72.2 | 67.7 |

*5.3. Discussion*

Since existing input data from potential EV users was used, one can assume adequate results for the model, especially if uncertainty is considered for the more volatile data, i.e., the EV daily energy needed, number of CCPs users, and energy required.

The problem formulation presumed that the SOC can not be calculated directly as a simplistic battery charging method. In real life, some calculation variations are needed in the battery charge. Nevertheless, the model is able to provide reliable and valid results for large simulations of electric vehicles on a grid scale. In addition, advanced SOC estimators must be considered by the EV Aggregator to prevent these sensitivity errors in individual vehicles in future.

**6. Conclusions**

This paper presents the formulation and development of the proposed methodology for regulation services, with and without the consideration of the V2G mode. An evaluation of the revenues of the EV aggregator was analyzed.

Moreover, the regulation services were applied to the Ecuadorean grid, which does not have a regulation market. For this purpose, the information from Spanish market was used as an input for the model.

This study has shown, that the EV aggregator has a substantial potential for providing regulation services in electricity markets, especially considering the use of the V2G mode. But, the model does not present significant benefits for regulation up, with and without the V2G mode, mainly because of the considered flexibility of the EV users. It is crucial to mention also that the V2G could create uncertainty in the users about the real use of the EV battery from the EV aggregator and it requires significant technical requirements during the installation of V2G equipment. In the case study of Ecuador, the effectiveness of the participation in regulation services for EV was demonstrated. First, the EV aggregator receive benefits for this participation. Then, the results prove that the Ecuadorean grid could avoid grid payments to foreign countries for regulation purposes.

In this work, the smart charging methodology was applied to a typical distribution grid. Thus, it could result crucial to develop a methodology in distribution grids with a high presence of high RES.

**Funding:** This research received no external funding.

**Acknowledgments:** The author wishes to thank Carlos Álvarez-Bel from Universitat Politècnica de València and Antonio-Marco Pantaleo from Universita degli Studi di Bari Aldo Moro, for providing important suggestions for this work.

**Conflicts of Interest:** The author declares no conflict of interest.

## Abbreviations

The following abbreviations are used in this manuscript:

| | |
|---|---|
| CCP | Customer Choice Product |
| EV | Electric Vehicle |
| LFC | Load Frequency Control |
| RES | Renewale Energy Source |
| V2G | Vehicle-to-Grid |

## Nomenclature

**Parameters**

$\alpha_P$ Penalty coefficient ($/kWh)

$\alpha_U$ Coefficient from the EV aggregator that is paid to the EV users who participate in ancillary services

$\alpha_{bd}$ Battery degradation cost due to discharging ($/kWh)

$\underline{\Delta T}$ Time interval

$\overline{P^{EV}}$ Maximum charge power rate for EVs (kW)

$\overline{SOC^{EV}}$ Maximum state-of-charge for EVs (%)

$\pi^{E,D}$ Market selling price for regulation down ($/kWh)

$\pi^{E,U}$ Market selling price for regulation up ($/kWh)

$\underline{P^{EV}}$ Maximum discharge power rate for EVs (kW)

$\underline{SOC^{EV}}$ Minimum state-of-charge for EVs (%)

$c_t^D$ Hourly regulation price down ($)

$c_t^D$ Hourly regulation price up ($)

$c_t^E$ Price of electricity at time $t$ ($)

$E^D$ Dispatched energy for regulation down (kWh)

$E^U$ Dispatched energy for regulation up (kWh)

$N^B$ Number of vehicles participating in blue CCP

$N^G$ Number of vehicles participating in green CCP

$N^R$ Number of vehicles participating in red CCP

$N_{EV}$ Total number of EV users

$r^D$ Dispatch ratio for regulation down

$r^U$ Dispatch ratio for regulation up

$R_t^D$   Regulation capacity down available at time $t$ (kW)

$R_t^U$   Regulation capacity up available at time $t$ (kW)

$R_t^{D,B}$   Offered regulation bid down for time $t$ (kW)

$R_t^{U,B}$   Offered regulation bid up for time $t$ (kW)

**Variables**

$C_B$   EV aggregator Costs from battery degradation ($)

$C_E$   EV aggregator Costs from cost of energy ($)

$C_H$   EV aggregator Costs from bi-directional chargers($)

$C_P$   EV aggregator Costs from penalty charge ($)

$E_P$   Energy not complied from the EV aggregator (kWh)

$P_t^{bas}$   Power prediction at time $t$

$P_t^{EV,d}$   Total power discharged from EVs (kW)

$P_t^{EV}$   Total power supplied to EVs (kW)

$P_{i,t}$   Charging power rate from EV $i$ at time $t$ (kW)

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
