# Peer review of "Participation of Electric Vehicle Aggregators in Ancillary Services Considering Users’ Preferences"

_sustainability, doi:10.3390/su12010008_

Round 1
Reviewer 1 Report
Please review the language - in general the paper reads well, but I noticed a couple of mistypes and less usual phrases.
Author Response
Thank you for your comments. The manuscript was proofread.
Reviewer 2 Report
This paper updates a previous study published by the author, where he devised a smart charging scheme for electric vehicles (EV) considering user preferences, by allowing the EV aggregator to participate in the market for ancillary services.
This paper addresses a very relevant topic, which well within the scope of the journal; it is reasonably well written and organized; methods are adequate and clearly explained; results are well illustrated and support the conclusions. Usage of the English language should be improved. However, I suggest to correct or clarify the following points.
Main issues
° The figures representing price profiles for regulation up and down (referenced at lines 302-303) are missing.
° In the Conclusions (line 355) the author states that "the Ecuadorean grid could avoid grid payments to foreign countries for regulation purposes". It is the first time that this problem of paying foreign countries for regulation services is mentioned, please add some details about it in the preceding sections.
° Eq. (2): The terms related to regulation capacity should be enclosed in brackets as both must be summed up. Furthermore, since time intervals are not exactly equal to 1 hour, regulation capacity terms should be multiplied by the time interval ΔT.
° All equations: the period is not a proper multiplication symbol! Use the multiplication sign ×.
° The authors uses "$/kWh" to denote energy price, and "$/kW-h" to indicate price coefficients for regulation capacity. Even though the quantities do indeed have different meanings (the former is the price of 1 kWh of energy exchanged with the grid; the latter the price for 1 kW of capacity made available for a period of 1 h), they should be represented by the same units. Furthermore, the symbol "kW-h" is not even a correct unit in the SI.
Minor issues
° Line 16, the sentence "In the future Smart Grid, users will become active prosumers" repeats what is already stated at line 15.
° The first time that the reader knows that the algorithm was tested on an Ecuadorean case study is at line 179, well within the Methods section, and with just a passing remark. I think it should be made clear in the Introduction.
° Eq. (19) contains the parameter pF: explain its meaning (I guess it is a "fixed price"?)
° Line 244, "Ecuador is considered again": for sure "again" refers to the case study presented in Ref. [13], but it should be made apparent with a bibliographic reference.
° Refs. [21], [28], [32] miss some bibliographic details, please update.
English language
The author should thoroughly review his paper to improve English language and style. Some examples of sentences that need attention are the following:
° In several cases, the author uses the subject after the verb, and places "it" before the verb. These sentences should rephrased with the subject preceding the verb, removing the redundant "it". For example:
line 116, "it was considered the minimization..."
line 229, "it will also be known the moment..."
line 316, "It is observed different variations..."
line 353, "it was demonstrated the effectiveness..."
° line 39, replace "do not considered" with "did not consider".
° line 43, replace "as follow" with "as follows".
° lines 85-86, "maintain" is used twice as a noun, which it is not: rephrase.
° line 117, "with the participation of it in te regulation services": correct "te" and rephrase the whole sentence.
° line 143, replace "bidirectionnal" with "bidirectional".
° line 156, replace "this costs" with "these costs".
° line 156, "they are the only that they can use the bidirectional chargers": rephrase.
° line 211, replace "exploit" with "exploitation" or similar.
° line 269, rephrase the sentence "For this study, it is assumed that different levels of energy are going to follow the pattern of this probability curve, but considering last average value."
° line 296, replace "Ecuador do not have" with "Ecuador does not have".
° line 297, replace "CENACE as to" with "CENACE has to".
Reviewer 3 Report
This paper presents a meaningful research subject on electric vehicle aggregators in terms of participation in ancillary services. The main contribution of this work is to investigate the electric vehicle aggregator potential profitability from providing regulation services in electricity markets. Then, the revenues of the electric vehicle aggregator are evaluated, and the regulation services were applied to the Ecuadorean grid. The objectives of the paper are clearly presented, and the results may be valuable to regulation services. The conclusions are really thought-provoking, representing a contribution to the state of the art in the field. However, the presentation can be improved. The reviewer will recommend this paper for publication only if all the comments are addressed sufficiently. The detailed comments are as follows.
1. The Abstract should be rephrased since it is hard to highlight more important information. Abstracts usually have at least one sentence per each: context and background, motivation, hypothesis, methods, results, conclusions.
2. An Introduction, please update the literature survey by referring to the most recent and relevant references. Please further supplement some research progress of EV aggregator, this can better highlight the advantages of ancillary services considering users’ preferences.
3. What is the Frequency Control in Section 2.2? What is the role of the EV system?
4. How does the participation in ancillary services effect on electric vehicle aggregators, please describe how to work?
5. The Spanish market is used as input data for the V2G mode. What are the characteristics of the data for the model?
6. Could the author add more discussions on the accuracy of the result? How could the author validate the result?
7. Can you briefly explain the technical obstacles if the research of this manuscript is to be applied in engineering?
Round 2
Reviewer 2 Report
Dear author,
I think that the changes to the manuscript and your response to my review adequately address my concerns.
Reviewer 3 Report
The authors addressed all comments and revised the paper accordingly. The revised paper can be accepted.